# The Chlorophyll Fluorescence Parameter F_v_/F_m_ Correlates with Loss of Grain Yield after Severe Drought in Three Wheat Genotypes Grown at Two CO_2_ Concentrations

**DOI:** 10.3390/plants12030436

**Published:** 2023-01-18

**Authors:** Søren Gjedde Sommer, Eusun Han, Xiangnan Li, Eva Rosenqvist, Fulai Liu

**Affiliations:** 1Department of Plant and Environmental Sciences, Crop Sciences, University of Copenhagen, Højbakkegaard Allé 13, DK-2630 Taastrup, Denmark; 2Sino-Danish Center for Education and Research, 380 Huaibeizhuang, Beijing 101400, China; 3CSIRO Agriculture and Food, P.O. Box 1700, Canberra, ACT 2601, Australia; 4Chinese Academy of Sciences, Northeast Institute of Geography and Agroecology, Changchun 130012, China

**Keywords:** F_v_/F_m_, photosynthesis, transpiration, chlorophyll fluorescence, stomata conductance, grain yield, harvest index, drought stress, elevated CO_2_

## Abstract

Three genotypes of wheat grown at two CO_2_ concentrations were used in a drought experiment, where water was withheld from the pots at anthesis until stomatal conductance (g_s_) dropped below 10% of the control and photosynthesis (A) approached zero. The genotypes had different leaf area (Gladius < LM19 < LM62) and while photosynthesis and shoot growth were boosted by elevated CO_2_, the water use and drying rate were more determined by canopy size than by stomatal density and conductance. The genotypes responded differently regarding number of fertile tillers, seeds per spike and 1000 kernel weight and, surprisingly, the largest genotype (LM62) with high water use showed the lowest relative decrease in grain yield. The maximum photochemical efficiency of photosystem II (F_v_/F_m_) was only affected on the last day of the drought when the stomata were almost closed although some variation in A was still seen between the genotypes. A close correlation was found between F_v_/F_m_ and % loss of grain yield. It indicates that the precise final physiological stress level measured by F_v_/F_m_ at anthesis/early kernel filling could effectively predict percentage final yield loss, and LM62 was slightly less stressed than the other genotypes, due to only a small discrepancy in finalising the drying period. Therefore, F_v_/F_m_ can be used as a proxy for estimating the yield performance of wheat after severe drought at anthesis.

## 1. Introduction

Wheat is, after maise, the second most produced cereal globally, it takes up the most significant proportion of cultivation area (14%), and it is a primary contributor of calories and protein globally [1]. To feed a growing global population, it is projected that demand for wheat will increase by approximately 50% by 2050 [2]. Meanwhile, anthropogenic activity is driving climate change due to primarily emissions of greenhouse gases [3]. Consequently, more frequent and more severe drought episodes are expected [4]. 

Wheat is best adapted to temperate climate regions, with optimal growth at around 18 °C day/13 °C night temperature [5] and where rainfall is non-limiting for growth, flowering and kernel filling [6,7]. It is well known that drought stress, particularly at anthesis, will reduce yield, partly due to shortening the life cycle of the crop and the availability of photosynthates for grain filling [8,9]. Earlier studies have demonstrated that the effect of high temperature on the maximum quantum efficiency of photosystem II F_v_/F_m_ in different wheat genotypes correlates with higher stomatal conductance (g_s_), causing increasing levels of transpiration (E) on leaf level [10]. Drought does not affect the reaction centre of photosystem II (PSII) until the stress becomes severe because of several protective mechanisms. When the light harvest exceeds the energy demand in the chloroplast, the excess light is dissipated as heat through non-photochemical quenching (NPQ). If the CO_2_ supply becomes limiting due to decreasing stomatal conductance, photorespiration acts as an alternative electron sink for the light reaction [11].The latter protects PSII from damage during drought, and therefore, F_v_/F_m_ is not a good indicator for detecting plant drought response at mild soil water deficit, but with increasing water deficit the response of F_v_/F_m_ will become evident [12,13,14].

Genetic variability may determine photosynthetic performance between different wheat genotypes. At growth stages around anthesis and through grain filling, the sustenance of photosynthesis correlates well with final grain yield. This means that yield optimisation is related to the ability of wheat plants to sustain green foliage around and after anthesis [15,16]. Such a link becomes more critical when plants suffer from abiotic stress, such as drought and combined drought and heat stress at the post-anthesis stage [17,18,19]. 

CO_2_ levels may very well be at 600 ppm by the end of the current century, and the worst case scenario projects a level of 1100 ppm [4]. We have chosen 800 ppm as an intermediate CO_2_ scenario. Elevated CO_2_ (eCO_2_) may have positive effects on yield in both well-watered and water-limited conditions compared to current ambient CO_2_ levels (aCO_2_). The increased yield potential in well-watered conditions is a consequence of an increase in the photosynthetic rate, which is in an almost linear manner to increasing CO_2_ concentration from 0 ppm to 600 ppm [20]. The increase in CO_2_ levels depresses photorespiration as high atmospheric CO_2_ concentration counteracts the disproportionate solubilization of CO_2_ to O_2_ in water as well as the relative preference for O_2_ at the RuBisCO catalytic site [21]. Due to these changes in carbon assimilation_,_ increasing levels of CO_2_ (to 550–900 ppm) are predicted to increase yields by 20–30% [22], and can potentially ameliorate the impacts of drought stress on wheat crops [23,24]. The amelioration of drought by eCO_2_ is attributed mainly to a decrease in g_s_ as intercellular CO_2_ (C_i_) concentration increases [22]. It is shown that leaf stomatal density has been adapted to varying CO_2_ levels over geological time. Due to this effect, leaves would have increased stomatal density leading to higher potential g_s_ during periods with low atmospheric CO_2_ levels. Meanwhile, periods with elevated atmospheric CO_2_ levels would favour the adaptation of leaves with lower stomatal density and lower maximum g_s_ [25]. Lower stomatal density is not mainly related to net photosynthesis, therefore, crop yield. Instead, it is associated well with whole-season water use efficiency (WUE) and can be a beneficial trait under water-limited conditions [26].

To ensure high and stable wheat yields in the changing climate, there is a need to understand the responses in physiological traits such as stomatal conductance and photosynthesis under drought conditions. In this study, the physiological responses of three different genotypes of wheat plants grown under drought were examined in a CO_2_-enriched environment by photosynthesis, F_v_/F_m_ and leaf pigment measurements. The data collected here were combined with yield data to analyse the relationship behind the stress-resilient morphological/physiological traits and yield. It was hypothesised that: (1) According to the heat screening experiment, LM62, which sustains higher F_v_/F_m_ compared to LM19 under heat stress, would achieve higher F_v_/F_m_ because of a higher photosynthetic rate. As a higher photosynthetic rate allows for sustained high stomatal conductance, and due to the probability that LM62 may sustain higher g_s_, we hypothesise that because LM62 will transpire more, LM62 pots will dry out faster. (2) eCO_2_ could mitigate the impact of drought stress on plant–water relations, leaf gas exchanges and grain yield.

## 2. Results

### 2.1. Leaf Water Potential

Midday leaf water potential (Ψ_leaf_) was severely affected by drought compared to control. There was a significant interaction between genotype and watering treatment (*p* < 0.001) (Table 1A, Figure 1). Gladius had the least negative Ψ_leaf_ compared to LM19 and LM62 under drought and eCO_2_, while LM19 had slightly less negative Ψ_leaf_ than LM62 under both control and drought conditions. The Ψ_leaf_ of Gladius was slightly less negative in eCO_2_ compared to aCO_2_ when exposed to progressive soil drying, but not for LM19 and LM62 (Figure 1).

### 2.2. Stomatal Density

Stomatal density did not differ significantly between either treatment or CO_2_ concentration, though there was a significant effect of genotype (*p* < 0.001, Table 1B). Gladius and LM62 had a significantly higher stomatal density than LM19, while there was no significant difference between Gladius and LM 62 (Figure 2).

### 2.3. A, g_s_, Ci, E and F_q_’/F_m_’

Net photosynthetic rate, A, increased significantly in control conditions as a response to eCO_2_ (*p* < 0.001, Table 1A). Compared to plants grown under aCO_2_, plants grown under eCO_2_ possessed 32%, 26% and 28% greater A (*p* < 0.05) for Gladius, LM19 and LM62, respectively. There was no significant effect of the genotype on A under control conditions (Figure 3A,B). 

There was a significant effect of the genotype on the gas exchange rates under drought (*p* < 0.001). LM62 had a higher A compared to Gladius and LM19 under drought (the A of LM62 was 84% lower at aCO_2_ and 78% lower at eCO_2_ in drought compared to control, while A was 93–99% lower in drought at both aCO_2_ and eCO_2_ for Gladius and LM19) (Figure 3A,B). A did not differ significantly between aCO_2_ and eCO_2_ in drought conditions.

Stomatal conductance, g_s_, was significantly lower (ca. 30%) at eCO_2_ compared to aCO_2_ in all genotypes under well-watered conditions (*p* < 0.001) (Table 1A, Figure 3C,D). This caused E to decrease to the same degree in eCO_2_ compared to aCO_2_ conditions (Figure 3C,D). C_i_ was significantly higher at eCO_2_ compared to aCO_2,_ even with the lower g_s_ (data not shown). When comparing g_s_ and E in drought, there was a significant difference between genotypes (*p* < 0.01). LM62 had a g_s_ of 32 mmol m^−2^ s^−1^, which was approximately 60% higher compared to Gladius (not significant) and 75% higher than LM19 (significant), which affected E with the same pattern as for g_s_ (Figure 3E,F).

F_q_’/F_m_’ did not differ significantly between either genotypes or CO_2_ conditions for well-watered plants (Figure 3G,H). However, in drought, the genotype had a significant effect on F_q_’/F_m_’ (*p* < 0.001) with significant interaction between the genotype and CO_2_ conditions (*p* < 0.001) (Table 1A). In drought, F_q_’/F_m_’ was significantly (50%) higher in LM62 compared to Gladius and LM19, with no effect of CO_2_ concentration.

### 2.4. F_v_/F_m_

F_v_/F_m_ was not significantly affected by CO_2_ but was significantly affected by genotypes (*p* < 0.05) and treatment (*p* < 0.001) as well as their interactions (*p* < 0.001) (Table 1A). Under aCO_2,_ drought caused a decrease of 9%, 14% and 3% in F_v_/F_m_ for Gladius, LM19 and LM62, respectively, compared to control. The reduction was significant for LM19 and Gladius (*p* < 0.05). Under eCO_2_, the decrease in F_v_/F_m_ by drought was 4%, 8% and 2% and only significant for LM19 (*p* < 0.05, Table 1A, Figure 4).

### 2.5. Pigments

The chlorophyll content (index) differed between genotypes (*p* < 0.01) and treatments (*p* < 0.001). Furthermore, there were significant interactions between the genotype, CO_2_ and treatment (*p* < 0.05, Table 1B). The chlorophyll content was significantly lower in Gladius and LM19 in drought conditions compared to the control (*p* < 0.05). The response was more significant in Gladius than in LM19. The chlorophyll content did not decrease in LM62 in drought compared to control conditions (Figure 5A,B). For Gladius and LM19 across all treatments, the chlorophyll content was slightly higher under eCO_2_ compared to aCO_2_ though not significant. In comparison, the chlorophyll content of LM62 was higher in aCO_2_ compared to eCO_2_ (though not significant).

The flavonol content (index) differed significantly between genotypes (*p* < 0.001). There was an interactive effect between the genotype and treatment on flavonol content (*p* < 0.05) (Table 1B). Under control conditions, the flavonol content was significantly higher (*p* < 0.05) in LM62 compared to Gladius and LM19 (not significantly in eCO_2_). Gladius and LM19 accumulated significantly more flavonols in response to drought compared to the control, which was not the case in LM62 (Figure 5C,D).

The anthocyanin content (index) differed significantly between control and drought (*p* < 0.001), and there was a significant interaction between CO_2_ and treatment (*p* < 0.001) (Table 1B). Anthocyanin levels were significantly higher in drought compared to control in Gladius and LM19 (*p* < 0.05). Meanwhile, the anthocyanin content of LM62 did not change significantly as a response to drought. In control conditions, the anthocyanin levels were significantly higher across all genotypes in eCO_2_ compared to aCO_2_, and for LM19 the difference was significant (*p* < 0.05) (Figure 5E,F).

### 2.6. Plant Size, Leaf Area and Water Use at Stress

Total dry weight (DW) and leaf area (LA) were significantly different between genotypes (Gladius < LM19 < LM62) (*p* < 0.001 and *p* < 0.05, respectively). Dry weight differed significantly between treatments and CO_2_ conditions (drought < control, aCO_2_ < eCO_2_) (*p* < 0.05), while LA was only affected significantly by treatment and not CO_2_ (*p* < 0.001). Water use (WU) and water use per leaf area (WU/LA) were similarly significantly different between genotypes (*p* < 0.001). While WU increased with increasing leaf area (Gladius < LM19 < LM62), WU/LA would decrease with increasing leaf area (LM62 < LM19 < Gladius), and LM62 had half of the WU/LA of Gladius (Table 2)), and eCO_2_ had a small but not significant interaction with the genotype on WU/LA, as WU/LA was lower in eCO_2_ compared to aCO_2_, which was most significant in Gladius and least significant in LM62 (*p* < 0.1) (Table 1B).

### 2.7. Yields

Grain yields were significantly different between genotypes (*p* < 0.001) and increased in accordance with the lateness of flowering, following the same trend as shoot dry matter (SDM) (Table 1C). There was no significant effect of eCO_2_ on yield. Meanwhile, in LM19 yield was significantly higher (20%) at eCO_2_ compared to aCO_2_ (Table 3). Drought affected grain yield significantly (Table 1C), causing a 70–90% (*p* < 0.001) decline in yields, which was in part due to a decrease in the number of spikes (SN) per plant. The number of spikes differed significantly between genotypes, and LM62 had significantly higher SN than Gladius and LM19. Drought significantly reduced SN by 43–70% (*p* < 0.001). The yield reduction caused by drought was also due to a reduction in grain number per spike (GNPS), primarily in LM62, i.e., a reduction of 60% at aCO_2_ and 90% at eCO_2_ compared to the control (*p* < 0.01). In Gladius and LM19, GNPS was also significantly reduced by drought, though not for LM19 at eCO_2_. Thousand kernel weight (TKW) was not significantly different between genotypes (*p* < 0.1), but there was a significant interaction between genotype and treatment (*p* < 0.05) (Table 1C). Drought caused TKW to decrease by approximately 50% in LM19 compared to control, whereas the decrease was approximately 30% for Gladius and 15% for LM62 with no significant effect of CO_2_. The number of spikes per tiller (Spikes/Tillers) was significantly affected by drought (*p* < 0.001) and genotype (*p* < 0.05), and there was a significant interaction between genotype and treatments (*p* < 0.001). Generally, the number of Spikes/Tillers was higher in control conditions and decreased less by drought in Gladius than in LM19 and LM62. The ratio was slightly higher in plants grown under eCO_2_ compared to under aCO_2_, but the interaction effect between treatment and CO_2_ was not significant (*p* < 0.1) (Table 1, Table 3).

## 3. Discussion

Gladius is known as a drought- and heat-tolerant genotype well suited to a Mediterranean climate [27], primarily attributed to its small canopy compared to northern varieties such as LM19 and LM62, which are bred for temperate climate regions with higher rainfall [28]. LM19 and LM62 are two Swedish varieties with contrasting fluorescence (F_v_/F_m_) responses to heat, with LM62 having higher F_v_/F_m_ after heat stress than LM19 (unpublished data). It was hypothesised that the differences between LM19 and LM62 would be related to the efficiency of leaf cooling by water loss through transpiration, where LM19 transpired less water than LM62. Based on this assumption, we also hypothesised that LM19 would be more suited to dry conditions than LM62 due to its more restricted water use. Furthermore, it was hypothesised that eCO_2_ would increase growth and yield in control conditions due to stimulated photosynthetic rates but also in drought conditions due to a lower stomatal conductance and transpiration rate, which would ameliorate the effect of drought. Gladius had the longest drying period among the three genotypes before reaching the end of the drought treatment (Appendix A) due to its smaller leaf area. Rewatering of the drought-treated pots was performed after 7 (LM62), 8 (LM19) and 11 days (Gladius) of withholding irrigation. All genotypes reached a similar mid-day leaf water potential at the end of drought stress apart from Gladius in eCO_2_ conditions (Figure 1), which somewhat confirms the first hypothesis, but not entirely when studying the morphological versus the physiological characteristics of each genotype (Table 2), as well as the effect drought had on yields (Table 3). 

Staying green is a vital trait for enduring drought, and a greenness index at both anthesis and especially in the senescence period was used to evaluate the drought tolerance of wheat plants from anthesis and forward [18]. From visual assessment, LM62 maintained a green canopy in drought-stressed plants one month after rewatering, compared to Gladius and LM19, which had reached maturity (Appendix A). This was accompanied by higher values of A, g_s_, E, F_q_’/F_m_’, F_v_/F_m_ and chlorophyll content in drought-stressed LM62 compared to LM19 and Gladius under both aCO_2_ and eCO_2_. Meanwhile, the duration of the period of drying before reaching a g_s_ < 10% of control was primarily attributed to the canopy size, where the smaller leaf area of Gladius extended the drying period compared to LM19 and LM62 (Table 1B), which rejects Hypothesis 1, that any physiological difference attributed to variation in genotype apart from leaf area would make LM19 more drought tolerant than LM62 even though SD was significantly lower in LM19 (Figure 2).

Other studies have tried relating the greenness index or “stay-green” trait to several QTL regions associated with a diverse group of wheat progeny [19]. While some evidence pointed to QTLs that were related to root angle, the primary trait associated with the stay-green trait and yield sustenance was co-located with the Rht-dwarfing genes even though they sought to control for this trait [19], supporting the results reported here, that the one genotype with a dwarf phenotype (Gladius) sustained the longest drying period. At the same time, the results supported the notion that staying green correlates well with minimising yield loss. Here, we define the ability to stay green as a consequence of sustaining green foliage one month after stress (Appendix A). 

A doubling of CO_2_ concentrations from 400 ppm to 800 ppm had a significant effect on g_s_ as it decreased by approximately 30% (Figure 3C,D), which has been shown in previous studies (e.g., [29]), and not far from a consensus determined through a meta-analysis approach [22]. However, while the decrease in g_s_ caused E to decrease correspondingly, it had a limited effect on the speed of soil drying, where only LM62 had lower water consumption in eCO_2_ compared to aCO_2_ in control conditions (Table 2). One explanation may be that the canopy size (LA and DW in Table 2) increased in eCO_2_ for all genotypes, which increased WU. Nonetheless, the water use per leaf area (WU/LA) still decreased in eCO_2_ compared to aCO_2_, coinciding with the response of g_s_ and E (Table 2). This, to some extent, confirms Hypothesis 2, that eCO_2_ ameliorates drought effects. Still, as plants grow bigger in eCO_2_, probably in part because our plants suffered no restrictions in nutrient availability, the effect was relatively small in terms of drought amelioration.

Under control conditions, the absolute WU primarily increased with increasing LA. In this experiment, the leaf area decreased in the order LM62 > LM19 > Gladius. Plants with a higher leaf area obtained a larger canopy volume (Appendix A). The microclimate within a canopy differs from the surroundings, where less turbulent air flow creates a higher air humidity [30]. The higher air humidity contributes to decreased VPD within the canopy, which limits transpiration and water use [31]. Therefore, the WU/LA increased as the leaf area decreased (Table 2) in the order Gladius > LM19 > LM62, which could explain why a larger canopy would transpire less on a leaf area level. However, because Gladius had the smallest leaf area, it still had the lowest WU of the three genotypes. 

Stomatal density (SD) did differ between genotypes, with LM19 having lower SD than Gladius and LM62 (Figure 2), but this did not alter the g_s_ under control conditions (Figure 3). As a consequence, the difference in SD did not result in any significant effect on WU in accordance with other studies [26,32]. Nevertheless, on a broader scale, SD differences have been subject to evolutionary pressure by CO_2_ concentrations and water availability [25,33]. Moreover, point mutations that restricted SD in barley (*Hordeum vulgare*) had significant positive effects on water use and harvest yield under drying conditions [34]. This suggests that evolutionary pressure has affected SD on larger timescales and that SD can affect transpiration. Still, our results do not indicate that this was the case in our setup.

F_v_/F_m_ is generally unaffected by drought until the final stages of the stress when loss of turgor approaches [12]. In winter wheat, it has been shown to occur only when the leaf relative water content (RWC) drops below ca 70% [14], and even then, the decrease in F_v_/F_m_ is limited. In this experiment, F_v_/F_m_ was maintained at the control level until A decreased below five µmol m^−2^ s^−1^ (Figure 3A) and F_q_’/F_m_’ below 0.11 (Figure 3D) on the last day of the stress. Then F_v_/F_m_ decreased significantly and only in plants where A was < 2 µmol m^−2^ s^−1^ and F_q_’/F_m_’ < 0.07 (cf. Figure 3A,B,G,H and Figure 4) and when the leaf water potential dropped below −2.5 MPa (Figure 1). The difference in F_v_/F_m_ between genotypes was caused by the different degrees of stress on the last day of drought treatment. In the control plants, F_v_/F_m_ was independent of both A and F_q_’/F_m_’ but decreased along with the decrease in the two parameters on the last day of stress (Appendix A). Even though the drought stress ended when all treatments reached g_s_ < 10% of the control, LM62 still maintained a higher A compared to the control than the other genotypes did. In LM62, F_v_/F_m_ was still on the level of the control at the end of the drought treatment. 

The primary determinant of the grain yield in the control plants was the length of the growth period, most notably the timing of flowering, which differed between the genotypes (Appendix A). The primary determinant is the shoot dry matter, which in this experiment was related to both the number of spikes and the ratio of spikes/tillers (Table 3). A CIMMYT study based on their elite cultivars found that the primary yield determinant was SDM and the height of plants [35], which coincides with the results in the present study. As expected, early flowering (Appendix A) and small canopy (Table 2, Appendix A) were the primary traits of Gladius in avoiding drought stress, as it took a longer time for A to approach zero compared to LM19 (3 days faster) and LM62 (4 days faster than Gladius). This result is in agreement with the results reported in a similar setup comparing Gladius to the high-yielding British elite cultivar Paragon [28]. In that study, the seed set was 30% of the control in Paragon compared to 70% in Gladius in the drought treatment, and the reduction in yield was 15% in Gladius and 20% in Paragon in drought compared to the control. This would confirm a general trend that drought-tolerant plants produce fewer spikes, but a higher percentage of spikes survive under drought stress, and they show a lower abortion/infertility of spikelets relative to control. A consequence of severe drought is that fewer photosynthates are provided to each spike, resulting in a lower TKW [29]. A lower number of tillers would produce a higher overall TKW when plants are subjected to drought as the sink is smaller. This is confirmed by the HI, which increases as the canopy becomes smaller and produces fewer tillers. We found that a lower number of tillers is an advantage under drought conditions as more spikes produce a successful yield, and fewer seeds is an advantage as the TKW increases, which tends to generate a higher overall yield [29]. Moreover, the yield reduction caused by drought was smaller in Gladius compared to LM19, while the grain yield of LM62 was the least affected by drought.

It was not the scope of this experiment to test the effect of the exact timing of drought stress but from Appendix A it is clear that while all the genotypes maintained green foliage at the end of the drought, one month after drought treatment, Gladius and LM19 were unable to support the growth of newly developed tillers while LM62 sustained such capacity. Also, based on the yield data (Table 3) and Appendix A, an increase in yield under eCO_2_, especially for LM62, was partly due to generally more biomass produced and partly due to more green area one month after stress. 

At first glance, our results showed differences in drought tolerance between the genotypes. The drought duration needed to reach g_s_ < 10% of the control differed between the genotypes because of their different leaf areas. This overruled any effect of differences in stomatal density and conductance. Even though g_s_ approached zero, it created a slightly different drought severity in the three genotypes, and with those low values, A was strongly g_s_ dependent. This severe drought caused a wide range of 67–90% loss in grain yield. 

Since F_v_/F_m_ is a poor parameter to detect early stages of drought (Appendix A) [12], it has mainly been used to detect the effects of heat stress in wheat [10,36]. In tomatoes, it has been shown that screening for heat tolerance by F_v_/F_m_ in young, vegetative plants in climate chambers can be extrapolated to the field [37]. In the present experiment, the per cent loss of grain yield followed the same pattern as F_v_/F_m_ after drought in the order LM62 < Gladius < LM19 with a clear correlation (r^2^ = 0.91) between the parameters at both CO_2_ concentrations (Figure 6). Such a correlation between F_v_/F_m_ and the loss of grain yield has consequences for how the difference of drought tolerance between the genotypes can be interpreted. When the drought stress was severe enough, and A gradually approached zero, F_v_/F_m_ decreased. The close correlation between the loss of grain yield and F_v_/F_m_ indicates that the genotypes had reached different levels of final stress before being rewatered. In this experiment, this difference in stress level rather than intrinsic differences between the genotypes caused the final loss of grain yield.

Only little emphasis has been given to the extrapolation of knowledge gained in greenhouse conditions to field conditions. Poorter et al. (2016) [38] found that indoor experiments are primarily source-limited due to lower DLI during growth, corresponding well to our experiment (DLI ~12 mol m^−2^ d^−1^) (Appendix A) while in field conditions growth is primarily linked to source limitation as temperatures are on average lower. The implication is that the relative growth rate is higher in greenhouse conditions, while the final biomass is higher in the field on an area basis due to a more extended growth period. Next, plant densities are much lower in pot experiments, and wind conditions are significantly different compared to growing in the field. Lower canopy densities within greenhouse settings limit the amount of competition between individuals; consequently, tiller numbers increase. Less wind leads to greater variability in light and humidity gradients between inside and outside the canopy. Given field conditions, flowering time and canopy height may still be essential variables if they contribute to biomass accumulation over the growth period until anthesis [35]. Meanwhile, since high tillering largely determined spike number and, thereby, yield on a per-plant basis, one could expect the relative difference between the three genotypes to be smaller in field conditions. While one could assume that SD could contribute to yield determination in the field, some evidence suggests that it is not the case for wheat [39], where the latter is in accordance with our findings. In this experiment, the response mechanisms of three genotypes to drought were investigated. The conclusion was that the loss of grain yield after severe drought correlated more to the final stress level (g_s_ < 10% of control, photosynthesis approaching zero and decreasing F_v_/F_m_) when the plants were rewatered than to the genotype. Under milder drought stress, it will not be possible to use F_v_/F_m_ to determine the precise stress level. However, the results still put a question mark on whether it is at all possible to phenotype genetic differences in drought tolerance in pot experiments when the precise drought stress level has such a profound effect on the results. 

## 4. Conclusions

Our results suggest that the primary factor determining the time it takes from the onset of drying until g_s_ < 10% of the control is the canopy size of each genotype rather than stomatal density or transpiration per leaf area. Meanwhile, we found that the canopy size affected plant transpiration, as a bigger canopy generally transpired more water but also possibly alleviated the effect of higher transpiration due to a higher relative humidity within the canopy. eCO_2_ ameliorated the impact of drought in terms of leaf level E and, to some extent, yield as well. Still, since eCO_2_ increased leaf area due to a higher A, the positive effect was relatively small, even when drought stress was less significant. To our knowledge, it is the first time a close correlation between F_v_/F_m_ and the loss of grain yield after severe drought has been shown. The results indicate that the final stress level was more important for the relative loss of grain yield than the genotype.

## 5. Materials and Methods

### 5.1. Plant Material

Three genotypes of spring wheat (*Tritium aestivum* L.) with an expected contrasting tolerance to heat and drought were used. The Australian genotype ‘Gladius’ is characterised by a small leaf area and early flowering, making it relatively resilient to dry conditions [27]. Two breeding lines from Lantmannen Seed (Svaloev, Sweden) were selected from a heat resilience screening (unpublished data) using the fluorescence parameter F_v_/F_m_ as described by Sharma et al. (2012). Based on the analysis, a heat-resilient (LM62) and a heat-sensitive (LM19) genotype were chosen. Apart from traits related to F_v_/F_m_ at high temperatures, flowering time would also differ between the genotypes. The genotypes had different developmental rates. Gladius flowered early and produced small plants. LM19 would flower at an intermediate time between Gladius and LM62, while LM62 would flower late and produce the biggest plants (Appendix A). Eight seeds were sown on 29 February 2020, in 4 L pots containing 1600 +/− 10 g nutrient-enriched peat substrate of uniform moisture (Krukvaextjord med lera och kisel, SW Horto AB, Hammenhoeg, Sweden). The pots were placed in two neighbouring greenhouse compartments under ambient (400 ppm) and elevated (800 ppm) CO_2_ concentrations, respectively. Two weeks after seeding each pot, each was thinned into two plants. One was marked for non-destructive physiological measurements and final harvest for grain yield, and the other for destructive analysis during the treatment period. When the plants were not exposed to the drought stress treatments, they were irrigated daily according to demand with a full nutrient solution of electric conductivity (EC) 2.0. mS cm^−1^ and pH 6.0 administrated as ebb-flow watering via the central watering/nutrient computer of the greenhouse (AMI Completa, Senmatic A/S, Soendersoe, Denmark). 

### 5.2. Growth Conditions and Treatments

The experiment contained a control group that was watered sufficiently throughout the growth period and a treatment group subjected to progressive soil drying at anthesis. The control growth conditions were 22.0 ± 0.7 °C/17.0 ± 0.4 °C day/night temperature (DT/NT) and 70% relative humidity (RH) in two compartments with ambient (aCO_2_, 400 ppm) and elevated (eCO_2_, 800 ppm) CO_2_ concentration, respectively. The growth conditions are shown for DT/NT, RH, air vapour pressure deficit (VPD), CO_2_ concentration and daily light integral (DLI) for the whole growth period (Appendix A). The air temperature was controlled by the greenhouse water-to-air heating system, insulation/shade screens and vents and active cooling by air-to-air conditioning when needed. 

Since the three genotypes had different developmental rates, each genotype was subjected to the drought treatment when it reached anthesis (growth stage of 50–60 on the BBCH scale [40,41]). The drought treatment was initiated by withholding irrigation from the pot until the mean stomatal conductance was <10% of the control and the mean net CO_2_ assimilation rate approached zero. Following the drought treatment period, drought stressed plants were returned to a regular irrigation schedule until maturity.

Before the onset of the experiment, four pots were watered to full pot water-holding capacity, and the mean weight of these pots was used as the pot weight at 100% water-holding capacity. Ninety per cent of the weight at water-holding capacity was used as the watering target for the control pots. Change in water content for the drought-treated plants was calculated as the fraction of transpirable soil water according to [42] with the equation:(1)$$\mathrm{FTSW}=\frac{\left({\mathrm{WT}}_{n}-{\mathrm{WT}}_{f}\right)}{\mathrm{TTSW}}$$
where FTSW is the fraction of transpirable soil water, WT_n_ is the current pot weight, WT_f_ is the final pot weight where net photosynthesis reached 0, and TTSW is the total transpirable soil water in the pot.

### 5.3. Leaf Water Potential

Midday leaf water potential (Ψ_leaf_) was measured on the last day of the stress by a Scholander-type pressure chamber (Soil Moisture Equipment Corp., Santa Barbara, CA, USA). If the pressure within the pressure bomb reached 3 MPa, pressurisation was stopped, as air would begin to leak from the rubber gasket.

### 5.4. Leaf Gas Exchange and Chlorophyll Fluorescence

At the end of the treatment, leaf gas exchange parameters, including net assimilation rate (A), stomatal conductance (g_s_), transpiration rate (E) and operating efficiency of PSII (F_q_’/F_m_’), were measured on flag leaves of plants from each treatment (*n* = 4) with a GFS-3000 infrared gas analyser (IRGA) fitted with leaf chamber 3010-GWK1 and the 10% blue/90% red LED array/PAM-Fluorometer 3056-FL (Walz, Effeltrich, Germany). The measurements were performed at a PPFD of 1500 µmol m^−2^ s^−1^, 25 °C cuvette temperature, 400 ppm CO_2_ concentration for aCO_2_ and 800 ppm for eCO_2_. Cuvette absolute humidity was kept at 20,000 ppm creating a leaf-to-air vapour pressure deficit (VPD) in the range of 0.8–1.2 Pa kPa^−1^.

### 5.5. Stomatal Imprints

At the end of the treatment period, stomatal imprints were taken from four plant replicates at both the adaxial (*n* = 4) and abaxial side (*n* = 4) of the flag leaf at growth stage BBCH = 70–75. The imprints were collected by application of a silicon impression material (Elite HD+, Zhermack, Badia Polesine, Italy) and subsequently transferred to microscopy slides with transparent nail polish, according to (Smith et al., 1989). From each imprint, three images (technical replicates) were taken with a Leica DM 750 stereomicroscope with the digital Leica Application Suite (Leica Microsystems, Wetzlar, Germany) at different parts of the imprint. Stomata numbers were counted using AI software based on Convolutional Neural Network (CNN) called RootPainter [43]. The software allows interactive training via a user-friendly interface that has been validated against numerous plant datasets such as root images [44], root nodule and biopore images [43]. We split the original imprints into four tiles with a target width of 900 pixels for a rapid training process. We trained a model for two hours on a Google Colab notebook-based GPU (https://colab.research.google.com/drive/104narYAvTBt-X4QEDrBSOZm_DRaAKHtA (accessed on 4 January 2021)), using a corrective annotation strategy, in which the user delineated false positives and negatives in real-time training. The model was used for segmenting stomata on the original imprints. The region properties were extracted as CSV files, from which size calibration and the number of stomata per image were obtained (see pictures pre- and post-annotation in Appendix A).

### 5.6. Modulated Chlorophyll Fluorescence

The maximum quantum efficiency of PSII (F_v_/F_m_) was measured with a PAM-2500 (Walz, Effeltrich, Germany) after 30 min of dark adaptation with dark clips [12]. All measurements were performed between 11:00 and 15:00 on the final day of treatment and conducted on flag leaves of the main tiller (*n* = 8).

### 5.7. Non-Destructive Pigment Measurements

The chlorophyll index, and the index of photoprotective anthocyanins and flavonols, were measured with a Dualex Scientific+ (Force-A, Orsay, France) (*n* = 8).

### 5.8. Harvest and Yields

At the end of the stress period, four replicates from control conditions and four replicates (two replicates for Gladius at drought and eCO_2_) from the drought treatment were harvested to calculate the means of leaf area (LA) and plant total dry weight (DW). Meanwhile, the weight of all pots was measured every day between 14:00 and 16:00. The mean daily plant water use (WU) of the well-watered pots was calculated between consecutive days over the treatment period. It was calculated as the difference between the pot weight after watering and the pot weight the day after. When all plant material reached maturity, the whole plants were harvested. The number of tillers and spikes was counted, and spikes were threshed in a Staatmeister (Kurt Pelz, Bad Godesberg, Germany), grain numbers were counted in a Contador seed counter (Pfeuffer, Kitzingen, Germany) and weighed on a PB3002-s/FACT scale (Mettler Toledo, Greifensee, Switzerland).

### 5.9. Statistics

The open source program R, version 1.0.153 [45] was used for statistics, and all data were analysed by ANOVA. First, ANOVA was applied to estimate differences between genotypes, water treatments and CO_2_ conditions, and interaction effects. Statistical analysis was performed on linear models with no contributing random effects, and the assumptions behind the model outputs were accounted for through visual confirmation of accordance with a normal distribution and non-disruptive unequal standard deviation between groups. ANOVA was used to estimate the relation between treatment and genotype to physiology and yield traits, and the Tukey’s honest significance test accounted for variation between multiple groups. Estimates and significance between treatments and genotypes were made with the packages multcomp [46] and various R-packages from the tidyverse-collection [47].

## Figures and Tables

**Figure 1 plants-12-00436-f001:**
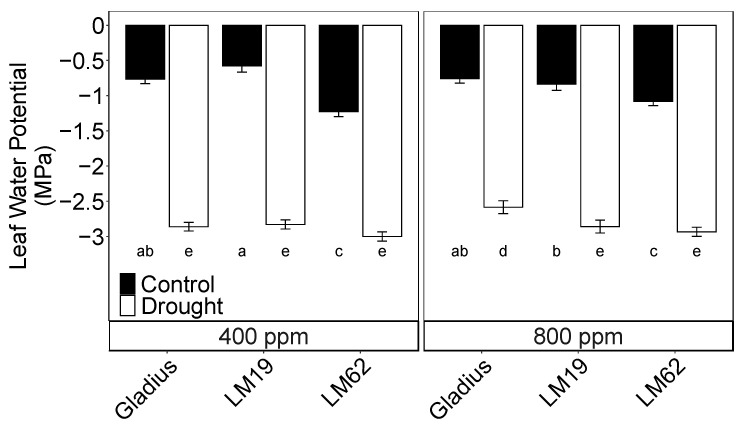
Midday leaf water potential of three spring wheat genotypes (Gladius, LM19 and LM62) under well-watered conditions (Control, black bars) or affected by drought (Drought, white bars) at aCO_2_ (400 ppm) and eCO_2_ (800 ppm). The measurements were conducted on a flag leaf around anthesis (BBCH = 65). Values are mean ± standard error of the mean (SE) (*n* = 4). Values that share a letter are not significantly different according to Tukey’s honestly significant test (*p* < 0.05).

**Figure 2 plants-12-00436-f002:**
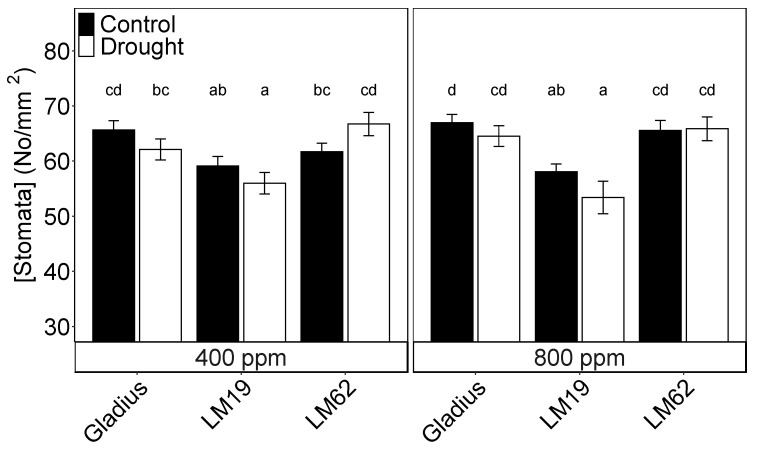
Stomata density for the three genotypes (Gladius, LM19 and LM62) under well-watered conditions (Control, black bars) or affected by drought (Drought, white bars) at aCO_2_ (400 ppm) and eCO_2_ (800 ppm), as a combination of adaxial and abaxial stomatal pore density. Measurements are taken at the end of the drying treatment on flag leaves. Values are mean ± standard error of the mean (SE) (*n* = 4). Values that share a letter are not significantly different according to Tukey’s honestly significant test. Small letters indicate a significant difference between a combination of watering, CO_2_, and genotype (*p* < 0.05).

**Figure 3 plants-12-00436-f003:**
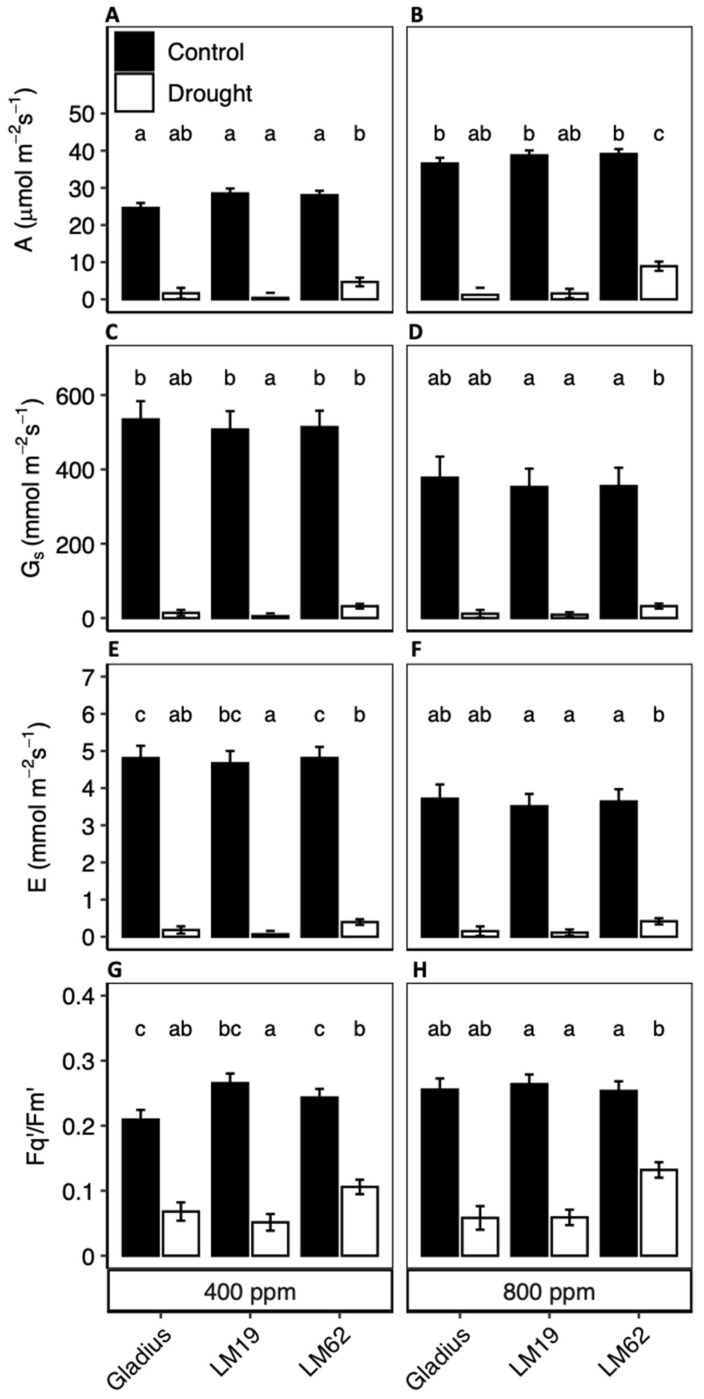
(**A**,**B**): Net photosynthetic assimilation (**A**), (**B**,**D**): Stomatal conductance (g_s_), (**E**,**F**): Leaf transpiration rate (**E**) and (**G**,**H**): quantum yield of PSII (F_q_’/F_m_’) of three spring wheat genotypes (Gladius, LM19 and LM62) under well-watered conditions (Control, black bars) or affected by drought (Drought, white bars) at aCO_2_ (400 ppm: (**A**,**C**,**E**,**G**)) and eCO_2_ (800 ppm: (**B**,**D**,**F**,**H**)). The measurements were done at a PPFD of 1500 µmol m^−2^ s^−1^ at the end of the treatment period. Values are mean ± SE (*n* = 4). Values that share a letter are not significantly different according to Tukey’s honestly significant test (*p* < 0.05).

**Figure 4 plants-12-00436-f004:**
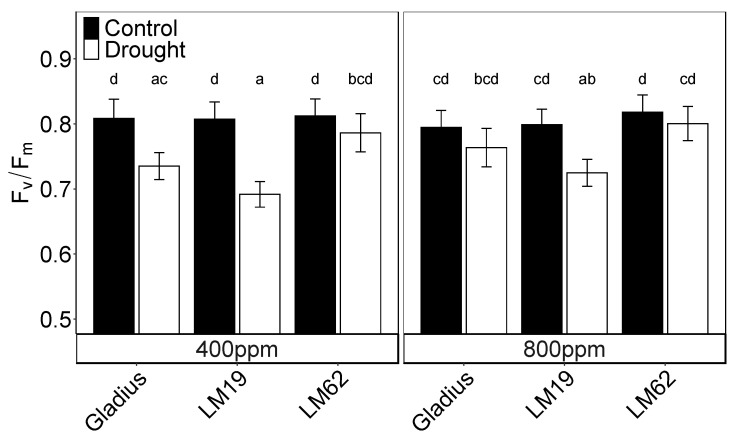
Maximum quantum efficiency of photosystem II (Fv/Fm) of the flag leaf of three spring wheat genotypes (Gladius, LM19 and LM62) under well-watered conditions (Control, black bars) or affected by drought (Drought, white bars) at aCO_2_ (400 ppm) and eCO_2_ (800 ppm). The measurements were conducted with a PAM-2500 on each genotype at the end of the treatment period. Values are mean ± SE (*n* = 8–16). Values that share a letter are not significantly different according to Tukey’s honestly significant test (*p* < 0.05).

**Figure 5 plants-12-00436-f005:**
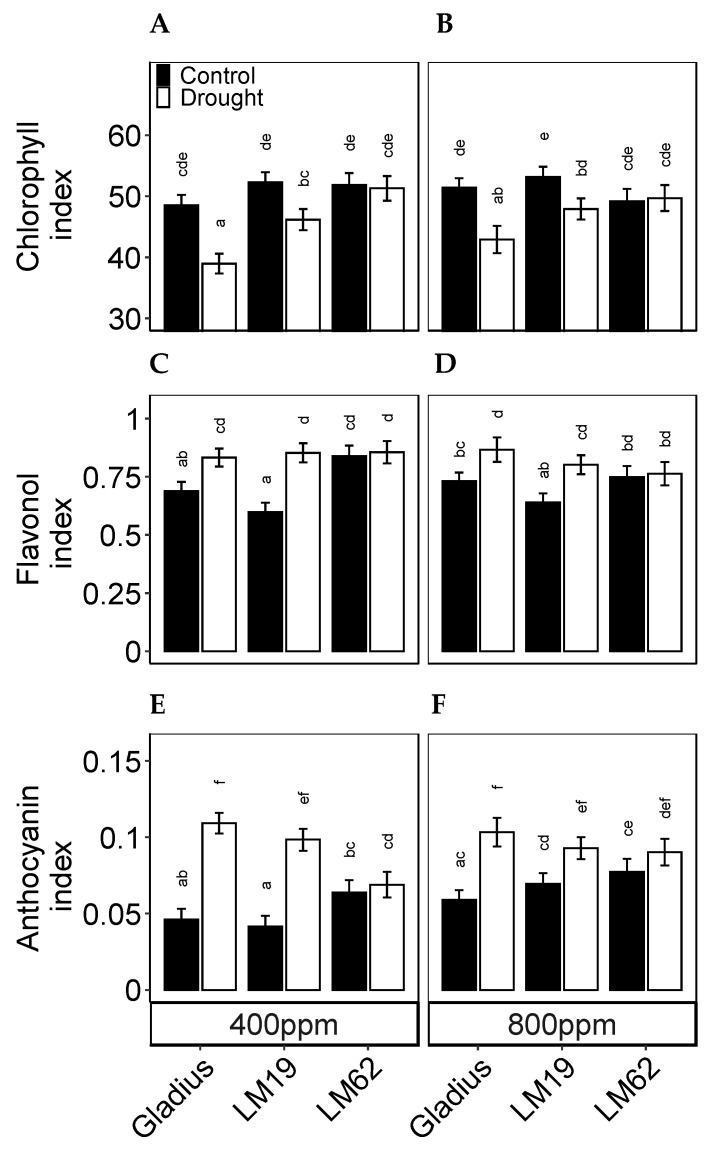
Pigment contents measured as Dualex indices. (**A**,**B**): chlorophyll, (**C**,**D**): flavonol and (**E**,**F**): anthocyanin index of flag leaf of three spring wheat genotypes (Gladius, LM19 and LM62) under well-watered conditions (Control) or affected by drought (Drought), at aCO_2_ (400 ppm: (**A**,**C**,**E**)) or eCO_2_ (800 ppm: (**B**,**D**,**F**)). Values represent mean ± SE (*n* = 8–16). Values that share a letter are not significantly different according to Tukey’s honestly significant test (*p* < 0.05).

**Figure 6 plants-12-00436-f006:**
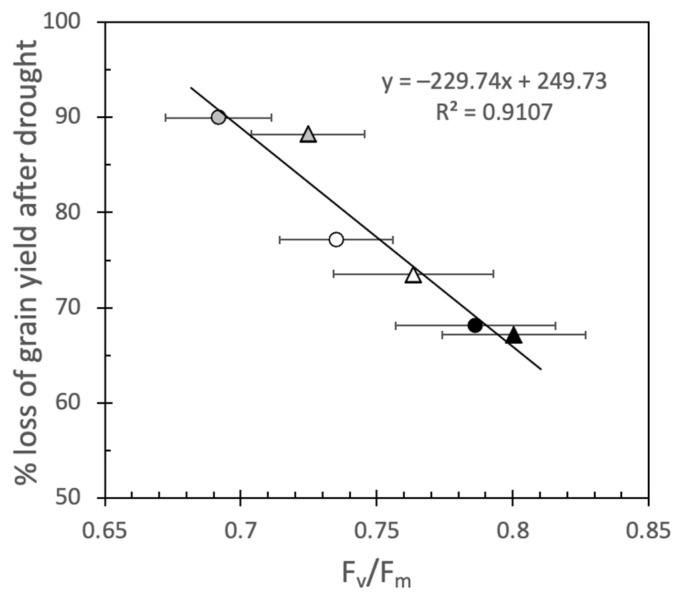
The relationship between loss of grain yield after severe drought and F_v_/F_m_ measured at the end of the stress period. Circles represent plants grown in aCO_2_, while triangles represent plants grown in eCO_2_. Open symbols represent Gladius, black represent LM62 and grey represent LM19. Values are mean ± SE (*n* = 8–16 for F_v_/F_m_ and % loss of grain yield calculated on mean values in Table 2).

**Table 1 plants-12-00436-t001:** Outputs of three-way ANOVA based on single factor effect and interaction effects divided into categories of A: Water status, gas exchange and chlorophyll fluorescence (Water potential, Net photosynthesis (**A**), stomatal conductance (gs), transpiration (E), PSII operating efficiency (Fq’/Fm’), Maximum quantum efficiency of PSII (Fv/Fm). Furthermore is the group of leaf properties and canopy characteristics at first harvest after drought stress (**B**): Stomatal density (SD), chlorophyll index (Chlorophyll), flavonol index (Flavonols), anthocyanin index (Anthocyanin), shoot dry matter (Harvest 1 SDM), leaf area (Harvest 1 LA), mean daily water use of control plants during the stress period (Harvest 1 WU), Mean daily water use per leaf area of control plants during the stress period (Harvest 1 WU/LA). The last table denotes (**C**) plant characteristics at final plant harvest: shoot dry matter (SDM), Grain yield, harvest index, number of spikes (SN), number of spikes per tiller (Spikes/Tillers), grain number per spike (GNPS), thousand kernel weight (TKW). The tests ANOVA were tested on the following variables: The three spring wheat genotypes Gladius, LM19 and LM62 (Genotype) as affected by drying conditions (Treatment) and CO_2_ conditions (CO_2_), along with interaction effects between the variables.

**A. Water Status, Gas Exchange and Chlorophyll Fluorescence**	**Water**	**A** **†**		**gs** **†**		**E** **†**		**Fq’/Fm’** **†**		**Fv/Fm**
	**Potential**	**Control**	**Drought**	**Control**	**Drought**	**Control**	**Drought**	**Control**	**Drought**	
Genotype	NS	NS	***	NS	**	NS	**	NS	***	*
CO_2_	NS	***	NS	***	NS	***	NS	NS	NS	NS
Treatment	***	***	–	***	–	***	–	***	–	***
Genotype + CO_2_	NS	*	***	NS	**	NS	**	NS	***	NS
Genotype + Treatment	***	–	–	–	–	–	–	–	–	*
CO_2_ + Treatment	NS	–	–	–	–	–	–	–	–	NS
Genotype + CO_2_ + Treatment	NS	–	–	–	–	–	–	–	–	NS
**B. Leaf Properties and First Harvest after Drought Stress**	**Stomatal**	**Chlorophyll**	**Flavonols**	**Anthocyanin**	**Harvest 1**	**Harvest 1**	**Harvest 1**	**Harvest 1**		
	**Density**				**SDM**	**LA**	**WU**	**WU/LA**		
Genotype	***	**	*	NS	***	*	***	***		
CO_2_	NS	NS	NS	NS	*	NS	NS	NS		
Treatment	NS	***	***	***	*	***	–	–		
Genotype + CO_2_	NS	NS	NS	NS	*	NS	.	.		
Genotype + Treatment	NS	***	*	NS	***	***	–	–		
CO_2_ + Treatment	NS	NS	NS	*	*	NS	–	–		
Genotype + CO_2_ + Treatment	NS	NS	NS	NS	**	–	–	–		
**C. Final Plant Harvest**	**SDM**	**Grain**	**Harvest**	**SN**	**Spikes/**	**GNPS**	**TKW**			
		**Yield**	**Index**		**/Tillers**					
Genotype	***	***	*	***	*	***	.			
CO_2_	NS	NS	NS	NS	NS	NS	NS	Statistical significance		
Treatment	***	***	***	***	***	**	***	***		*p* ≤ 0.001
Genotype + CO_2_	NS	NS	NS	NS	NS	NS	NS	**		*p* ≤ 0.01
Genotype + Treatment	***	***	***	***	***	***	*	*		*p* ≤ 0.05
CO_2_ + Treatment	NS	***	NS	NS	NS	NS	NS	.		*p* ≤ 0.1
Genotype + CO_2_ + Treatment	**	NS	NS	*	.	NS	NS	NS (not significant)		*p* > 0.1

† Data from water and drought treatments are divided into different ANOVA analysis of A, gs, Ci, as an ANOVA based on pooled data would obstruct the idea of data normal distribution.

**Table 2 plants-12-00436-t002:** Shoot dry matter (SDM), Leaf area (LA), from well-watered conditions (Control) and plants undergoing drought (Drought) at aCO_2_ (400 ppm) or eCO_2_ (800 ppm). Also, mean water use (WU) was calculated based on the daily water use of each control group during the treatment period. The mean water use per leaf area of each genotype in control conditions was calculated based on the daily water use divided by the total canopy leaf area. Values are mean ± SE (*n* = 4). Values that share a letter are not significantly different according to Tukey’s honestly significant test (*p* < 0.05). Values are mean ± SE (*n* = 4).

Genotype	CO_2_	Treatment	DW		LA		WU		WU/LA	
	ppm		g		cm^2^		g		g/cm^2^	
Gladius	400	Control	11.1 ± 3.9	a	758 ± 147	bc	251 ± 22	a	0.33 ± 0.02	d
		Drought	10.9 ± 3.9	a	134 ± 147	a				
	800	Control	20.5 ± 3.9	ab	1160 ± 147	c	235 ± 26	a	0.20 ± 0.02	bc
		Drought	23.7 ± 5.5	abc	187 ± 208	a				
LM19	400	Control	26.8 ± 3.9	bc	1881 ± 147	d	485 ± 30	b	0.26 ± 0.03	c
		Drought	21.7 ± 3.9	ab	223 ± 147	a				
	800	Control	40.9 ± 3.9	d	2558 ± 147	e	447 ± 24	b	0.17 ± 0.02	ab
		Drought	23.3 ± 3.9	b	346 ± 147	ab				
LM62	400	Control	36.4 ± 3.9	cd	3052 ± 147	f	510 ± 26	b	0.17 ± 0.02	ab
		Drought	25.6 ± 3.9	bc	362 ± 147	ab				
	800	Control	45.1 ± 3.9	d	3391 ± 147	f	454 ± 26	b	0.13 ± 0.02	a
		Drought	25.0 ± 3.9	b	720 ± 147	b				

**Table 3 plants-12-00436-t003:** Yield parameters: Shoot dry matter (SDM) (g), Grain yield (GY), loss in grain yield due to drought relative to control within the same CO_2_ conditions (GY loss), Harvest index (HI), spike number (SN), ratio of spikes to tillers (Spikes/Tillers), grain number per spike (GNPS) and 1000 kernel weight (TKW) of the three spring wheat genotypes (Gladius, LM19 and LM62) in control and as affected by drought under ambient (400 ppm) or elevated (800 ppm) CO_2_ conditions. Values that share a letter are not significantly different according to Tukey’s honestly significant test (*p* < 0.05). Values are mean ± SE (*n* = 4).

Genotype	CO_2_, ppm	Treatment	SDM, (g/plant) ± SE		GY, (g) ± SE		GY Loss, %	HI, (Ratio) ± SE		SN, (n) ± SE		Spikes/Tillers, (Ratio) ± SE		GNPS, (n) ± SE		TKW, (g) ± SE	
Gladius	400	Control	38.5 ± 5.2	b	17.7 ± 2.9	b		0.46 ± 0.03	g	23 ± 2.5	def	0.99 ± 0.04	d	26.0 ± 3.1	acd	32.2 ± 2.9	ce
		Drought	14.4 ± 5.7	a	4.1 ± 3.1	a	77%	0.27 ± 0.03	cd	7 ± 2.8	a	0.77 ± 0.05	b	22.6 ± 3.4	ac	23.7 ± 3.2	ac
	800	Control	41.7 ± 6.4	b	20.2 ± 3.5	b		0.48 ± 0.04	g	17 ± 3.1	bcd	0.95 ± 0.05	cd	30.9 ± 3.8	ce	38.2 ± 3.5	e
		Drought	15.6 ± 5.7	a	5.3 ± 3.1	a	74%	0.34 ± 0.03	de	9 ± 2.8	ab	0.91 ± 0.05	cd	25.6 ± 3.4	acd	26.0 ± 3.2	bcd
LM19	400	Control	73.2 ± 5.2	c	31.2 ± 2.9	c		0.43 ± 0.03	fg	23 ± 2.5	def	0.92 ± 0.04	cd	39.0 ± 3.1	e	34.7 ± 2.9	e
		Drought	29.7 ± 6.4	ab	3.1 ± 3.5	a	90%	0.11 ± 0.04	a	7 ± 3.1	a	0.47 ± 0.05	a	34.8 ± 3.8	de	14.8 ± 3.5	a
	800	Control	88.7 ± 5.2	d	38.9 ± 2.9	c		0.44 ± 0.03	fg	28 ± 2.5	f	0.90 ± 0.04	cd	39.7 ± 3.1	e	35.1 ± 2.9	e
		Drought	36.3 ± 6.4	b	4.6 ± 3.5	a	88%	0.13 ± 0.04	ab	13 ± 3.1	ac	0.61 ± 0.05	a	24.8 ± 3.8	acd	16.7 ± 3.5	ab
LM62	400	Control	128.1 ± 5.2	e	47.7 ± 2.9	d		0.37 ± 0.03	ef	46 ± 2.5	g	0.87 ± 0.04	bc	29.7 ± 3.1	bcd	35.6 ± 2.9	e
		Drought	66.0 ± 6.4	c	15.2 ± 3.5	b	68%	0.20 ± 0.04	ac	19 ± 3.1	ce	0.54 ± 0.05	a	19.9 ± 3.8	ab	28.6 ± 3.5	ce
	800	Control	149.8 ± 5.7	f	50.7 ± 3.1	d		0.34 ± 0.03	de	57 ± 2.8	h	0.91 ± 0.05	cd	24.0 ± 3.4	ac	36.5 ± 3.2	e
		Drought	78.2 ± 5.7	cd	16.6 ± 3.1	b	67%	0.21 ± 0.03	bc	27 ± 2.8	ef	0.61 ± 0.05	a	18.4 ± 3.4	a	33.0 ± 3.2	de

## Data Availability

For data inquiry, contact Fulai Liu or Eva Rosenqvist.

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
