# Peer review of "The Chlorophyll Fluorescence Parameter Fv/Fm Correlates with Loss of Grain Yield after Severe Drought in Three Wheat Genotypes Grown at Two CO2 Concentrations"

_plants, 2023, doi:10.3390/plants12030436_

Round 1
Reviewer 1 Report
This article compares a series of physiological indexes of three genotypes of wheat under drought and carbon dioxide treatment. Among them, Fv/Fm index is very distinctive and well written, which is suitable for this journal. My comments are as follows:
Major:
In the Introduction, it is suggested to explain the concentration range of CO2 in natural environment and the basis for choosing these two CO2 concentration treatments.
The full name of aCO2 and eCO2 should be represented when they were mentioned firs time. Line 88 and Line 98
Line 393. Figure 6. Author should explain the represents of triangle and dot. Also label three genotypes in this figure.
English writing needs to be improved.
There are many small mistakes such as punctuation marks.
Minor:
Line 74. The article is redundant, so it is suggest to delete "the" before "crop".
Line 77. Between climate and there, you need to add a comma.
Line 105,165,180. Change Tukeys to Tukey’s.
Line 127. Add “genotypes” after “for the three”.
Line 130. Delete “)” after “flag leaves”.

Author Response
Dear reviewers.
We thank you very much for reading through our work, and appreciate the feedback. We have looked through your comments, and found that all have merit, so we have made all changes accordingly. To clarify our final point about the effect of the stress level at the end of the experiment we have also updated the last figure so that all genotypes and growth CO2 concentrations can be identified for the drought stressed plants.
Reviewer 2 Report
In figure 5, no units were labeled in the Y-axis.
The authors concluded that the final stress level was more important for the relative loss of grain yield than the genotype. How to quantify the final stress level?
Author Response

(The authors gave the same response as above.)
